# Atherosclerosis Development and Aortic Contractility in Hypercholesterolemic Rabbits Supplemented with Two Different Flaxseed Varieties

**DOI:** 10.3390/foods10030534

**Published:** 2021-03-04

**Authors:** Jolanta Bujok, Dorota Miśta, Edyta Wincewicz, Bożena Króliczewska, Stanisław Dzimira, Magdalena Żuk

**Affiliations:** 1Department of Animal Physiology and Biostructure, Wroclaw University of Environmental and Life Sciences, C.K. Norwida 31, 50-375 Wrocław, Poland; dorota.mista@upwr.edu.pl (D.M.); edyta.wincewicz@upwr.edu.pl (E.W.); bozena.kroliczewska@upwr.edu.pl (B.K.); 2Department of Pathology, Wroclaw University of Environmental and Life Sciences, C.K. Norwida 31, 50-375 Wrocław, Poland; stanislaw.dzimira@upwr.edu.pl; 3Department of Genetic Biochemistry, Faculty of Biotechnology, University of Wroclaw, Przybyszewskiego 63/77, 51-148 Wrocław, Poland; magdalena.zuk@uwr.edu.pl

**Keywords:** aortic contractility, cholesterol, rabbit model, atherosclerosis, *Linola usitatissimum* L., genetically modified flaxseed

## Abstract

Alpha-linolenic acid (ALA) is widely regarded as the main beneficial component of flax for the prevention of cardiovascular disease. We evaluated the effect of the transgenic flaxseed W86—which is rich in ALA—on the lipid profile, atherosclerosis progression, and vascular reactivity in hypercholesterolemic rabbits compared to the parental cultivar Linola with a very low ALA content. Rabbits were fed a basal diet (control) or a basal diet supplemented with 1% cholesterol, 1% cholesterol and 10% flaxseed W86, or 1% cholesterol and 10% Linola flaxseed. A high-cholesterol diet resulted in an elevated plasma cholesterol and triglyceride levels compared to the control animals. Aortic sections from rabbits fed Linola had lower deposits of foamy cells than those from rabbits fed W86. A potassium-induced and phenylephrine-induced contractile response was enhanced by a high-cholesterol diet and not influenced by the W86 or Linola flaxseed. Pretreatment of the aortic rings with nitro-L-arginine methyl ester resulted in a concentration-dependent tendency to increase the reaction amplitude in the control and high-cholesterol diet groups but not the flaxseed groups. Linola flaxseed with a low ALA content more effectively reduced the atherosclerosis progression compared with the W86 flaxseed with a high concentration of stable ALA. Aorta contractility studies suggested that flaxseed ameliorated an increased contractility in hypercholesterolemia but had little or no impact on NO synthesis in the vascular wall.

## 1. Introduction

Cardiovascular disease accounts for 45% of deaths in Europe and 32% of deaths worldwide [1,2]. Atherosclerosis is considered a leading cause of cardiovascular mortality in developed countries [3]. A high-fat diet rich in saturated fatty acids (SFAs) and cholesterol as well as a low intake of fiber and polyunsaturated fatty acids (PUFAs) are recognized factors associated with atherosclerosis, which lead to recommendations of diets rich in PUFAs and poor in SFAs [4]. However, the impact of dietary fatty acids on atherosclerosis still remains controversial. Recent studies indicated no correlation between the consumption of SFAs and the overall mortality and also showed that some diets containing SFAs, such as dairy products, may be associated with a reduction in cardiovascular disease (CVD) risk [5,6,7]. Although the positive influence of PUFAs on CVD risk reduction appear to be obvious, some controversies appeared concerning the impact of the n-6/n-3 ratio. Generally, it is recommended to consume large amounts of n-3 and give less importance to n-6; however, certain studies involving humans stated the important roles of both n-3 and n-6 fatty acids, with no correlation between the n-6/n-3 ratio and CVD risk [8].

Flaxseed is a rich source of PUFAs and other active substances [9]. Standard flaxseed varieties contain up to 55% alpha-linolenic acid (ALA), representing the richest plant source of omega-3 acids. ALA is considered one of the main cholesterol-lowering and anti-atherosclerotic ingredients [10] and was shown to suppress the production of proinflammatory interleukins and lower C-reactive protein and serum amyloid-A concentrations in human patients and experimental animals [11]. However, in contrast to those previous research, current studies found that ALA exhibited very low anti-inflammatory properties, although they could be more pronounced due to its metabolic conversion to very long-chain PUFAs [12].

Although plant-derived ALA may be converted into eicosapentaenoic acid (EPA) or docosahexaenoic acid (DHA) via elongation and desaturation, these enzymatic reactions are inefficient in the human body [6]. Guichardant et al. [13] suggested the reducing effect of ALA on CVD risk was less evidenced than in case of marine-derived long-chain PUFAs—EPA and DHA. ALA appeared much less effective than EPA in the inhibition effect on the oxidation of related to atherosclerosis small dense lipoproteins (LDL) and membrane cholesterol domains [14]. Studies considered similar to ALA importance of the intake of linoleic acid (LA), despite no imbalance in the n-6/n-3 ratio [15].

Nevertheless, some studies confirmed the beneficial impact of flaxseed on CVD, e.g., consuming flaxseed by patients with coronary artery disease reduced cardiometabolic and inflammation biomarkers [16]. Therefore, research postulated that other substances with antioxidant properties, such as lignans, were responsible for the positive effects of supplementing flax in the diet [17]. The beneficial effect of flaxseed intake on CVD markers may be related not only to ALA, but to the combined actions of PUFAs with other plant components, such as fiber, polyphenols, and bioelements [15].

The ALA content and other bioactive components in flax seeds may depend on the variety. The transformation of *Linola usitatissimum* L. cv. Linola with a very low ALA content with the chalcone synthase gene controlling the phenylpropanoid pathway leads finally to increases in the ALA content. The immediate effect of modification is the accumulation of antioxidative substances, like proanthocyanids and hydrolysable tannins, in the seeds. 

The modified flaxseed W86 is, thus, rich in more stable ALA and antioxidants, which might affect the antiatherogenic activity of the plant [18,19,20]. Previous studies demonstrated anti-inflammatory and cholesterol-lowering effects of a transgenic flaxseed cake that was rich in bioactive polyphenols consumed by mice and rats [21,22,23]. The study concerning W86 indicated a decrease in the atherogenic indexes and atherogenic predictors in hipercholesterolemic rabbits fed W86 flaxseed, despite only a negligible effect on the serum lipid profile [24], which led us to make an attempt to estimate the effect of W86 seeds on the development of atherogenic lesions. 

The aims of this study were to determine if the Linola flaxseed with a very low ALA content influenced the serum lipid profile, atherosclerosis progression, and vascular reactivity in hypercholesterolemic rabbits and elucidate if an increase in stable ALA content in seeds obtained by genetic modification would further impact these effects. Since aortic stiffness is considered to be the earliest detectable manifestation of detrimental structural and functional changes within the aortic wall [25,26], the arterial contractile reactivity measurements may give some important information about the current status and functioning of the arterial wall, depending on the progress of atherogenic changes.

## 2. Materials and Methods

### 2.1. Animals and Experimental Design

The experiment was performed in accordance with the standards established in the directive of the EU (2010/63/EU) and in accordance with the principals of Replacement, Reduction, and Refinement (3Rs), and all procedures were approved by the Second Wroclaw Local Ethics Committee for Animal Experimentation (no. 60/2013). Twenty-eight male New Zealand White rabbits (Rabbit Breeding Unit, Małujowice, Poland) aged 6 months and weighing approximately 4 kg were used in the study. 

The animals were housed individually at 20 °C in cages with free access to water and a standard rabbit diet for 3 weeks of acclimatization period (G-103 pelletized mix for rabbits; Granum Animal Nutrition, Lubanow, Poland). Then, the animals were divided into four groups (*n* = 7 each) and fed different diets for 11 weeks. The control animals (C) received the basal diet used during the adaptation period. A group of animals was fed a 1% cholesterol-enriched basal diet and was named the cholesterol group (CH). Two groups of rabbits were fed the same 1% cholesterol-enriched diet containing 10% transgenic flaxseed W86 (group W) or 10% flaxseed Linola, (group L). 

### 2.2. Plant Material

The cultivar Linola (*Linola usitatissimum* L.) and transgenic W86 flaxseeds were obtained from the Department of Genetic Biochemistry, Faculty of Biotechnology, University of Wroclaw, where they were cultivated and where the W86 plants were generated. For W86 plant transformation, the cDNA encoding Petunia hybrida chalcone synthase (CHS database, EMBL/GenBank No. X04080), the 35S promoter and the OCS terminator were used [18]. 

Transgenic plants were selected by detecting the presence of the introduced gene (chs) and the expression of the endogenous flax chalcone synthase gene using the PCR technique. The genomic DNA isolated from tissue-cultured plants was used as a template. The selected plants showed repression of both introduced and endogenous chalcone synthase (co-suppression phenomenon). The Linola and transgenic plants were grown on a semi-technical scale in a field vicinity of Wroclaw (Poland) and the seeds were obtained after 3 months of growth. The obtained plants were characterized by a change in the color of the seeds in relation to the initial variety (Linola), which was most likely the result of the accumulation of hydrolysable tannins and proanthocyanides in the seed covers [27]. The increased accumulation of hydrolysable tannins was the result of the partial redirection of substrates from flavonoid biosynthesis (lower activity of chalcone synthase) to other pathways of the phenylpropanoid pathway—which increased the amount of gallic acid of the substrate for the production of hydrolysable tannins. 

These compounds have the ability to bind iron, which is an activator of fatty acid desaturase (FAD3) [18], and consequently the transgenic plants produced more (20%–45%) polyunsaturated fatty acids compared with the control, mainly α-linolenic acid with better stability due to the increased antioxidant capacity (Table 1). The details concerning plant transformation, selection, and transgenic plant analysis as well as the chemical characteristics of the W86 flaxseed were described in detail previously [18]. The fatty acid composition, as well as metabolites from the phenylopropanoid pathway contents of Linola and W86 flaxseed are presented in Table 1 and Table 2.

### 2.3. Blood Sampling and Analysis

After 11 weeks, the animals the animals were bled, and the plasma was isolated by centrifugation at 3000× *g* for 10 min, 4 °C. The total plasma cholesterol (TChol), high-density lipoproteins (HDL), LDL, and triglycerides (TG) were determined using an automated analyzer Pentra 400 (Horiba ABX, Kyoto, Japan) and dedicated reagents (Horiba ABX, Kyoto, Japan).

### 2.4. Tissue Preparation

After bleeding, the rabbits were euthanized by an overdose of intravenous sodium pentobarbital after sedation with a xylazine injection. Immediately following the sacrifice, the thoracic aortae were excised. Aortic fragments for in vitro contractile studies were immediately placed in cold Krebs–Henseleit solution (118 mM NaCl, 4.7 mM KCl, 2.5 mM CaCl_2_, 1.6 mM MgSO_4_, 24.3 mM NaHCO_3_, 1.18 mM KH_2_PO_4_, and 5.6 mM glucose; pH 7.4), carefully cleaned of adventitia, and divided into 4-mm-wide rings. The aortae for the histopathological studies were placed in 7% buffered formalin and stored at 20 °C until slide preparation. 

### 2.5. Histopathology

Aortic fragments were embedded in paraffin blocks, cut into 4-µm-thick transverse sections, and placed on glass slides. The preparations were stained with the routine hematoxylin and eosin (H&E) method and viewed using an Olympus BX 53 light microscope coupled with a camera model UC90. To take measurements, the cellSens Standard V1 software was used (Olympus, Tokyo, Japan). The thickness of the foamy cell depositions was evaluated as described previously by Brant et al. [28] and El-Sheakh et al. [29].

### 2.6. Aorta Contractile Studies

Thoracic aortae rings were mounted in 20-mL organ bath chambers filled with Krebs–Henseleit solution and bubbled with a 95% O_2_ and 5% CO_2_ gas mixture. The tissues were placed under 1-g resting tension and allowed to equilibrate for 60 min. The aortic contractions were recorded by isotonic transducers (Letica Scientific Instruments, Barcelona, Spain) connected to bridge amplifiers (BridgeAmp; ADInstruments, Dunedin, New Zealand) and a data acquisition system (PowerLab; ADInstruments, Dunedin, New Zealand) [30].

After the equilibration period, the tissues were stimulated with 60 mM KCl to determine depolarization mediated contractions. After a washout, 10 or 100 µM of nitric oxide synthase inhibitor nitro-L-arginine methyl ester (L-NAME) was added to some of the chambers, n samples specified where applicable. Following 15 min of incubation, a cumulative concentration–response curve to phenylephrine (PHE) was constructed to measure the α1-adrenergic mediated contractility. 

The potency of the PHE was determined by the curve fitting with non-linear regression and the derivatization pK value, which is the logarithm of PHE concentration that produces 50% of the maximum response. The efficacy of the agonist was expressed as the maximum response (E_max_) to the highest concentration of the agonist still producing an increase in tension. Moreover, the magnitudes of the contractile responses to increasing concentrations of PHE were compared by one-way ANOVA and post hoc Tukey test and the differences were shown on the graphs. To examine the relaxation response to sodium nitroprusside (SNP)—a nitric oxide donor, the aortic rings were precontracted with 10^−6^ M PHE, and cumulative concentration–response curves to SNP were constructed. E_max_ was expressed as percent reaction to 10^−6^ M phenylephrine elicited by the maximum concentration of SNP. Tissues that failed to produce tension in response to KCl or PHE were discarded.

### 2.7. Data Analysis

The data are expressed as the mean ± standard error of the mean (S.E.M.) of n samples. Sets of data obtained from the vascular contractile studies were fitted to dose–response sigmoidal functions with non-linear regression analysis using OriginPro 8 software. The potencies of the substances (pK) were derived from the obtained functions.

The data were subjected to one-way analysis of variance followed by post hoc Tukey’s test using the Statistica for Windows ver. 10.0 software package (StatSoft, Tulsa, OK, USA). Differences between the means were considered significant at values of *p* ˂ 0.05.

## 3. Results

### 3.1. Plasma Lipid Profile

Feeding a high-cholesterol diet for 11 weeks resulted in elevated blood plasma TChol and LDL levels. Neither W86 flaxseed nor Linola flaxseed supplementation lowered the plasma TChol and LDL concentrations. The HDL and TG concentrations remained unchanged in cholesterol-feed animals (Table 3).

### 3.2. Histopathology

In the control group, the normal structure of each layer of the aortic wall was preserved. The high-cholesterol diet resulted in a significant foamy cell deposition within the walls of the thoracic aorta. The addition of transgenic flax did not prevent the formation of atherogenic deposits, and their thickness was comparable to the changes in the CH group. On the other hand, the addition of the parental Linola cultivar was associated with the tendency to decrease the thickness of foamy cell deposits, and the thickness of the plaque in the aortic wall did not differ significantly from the control group, as opposed the aortae of the CH and W rabbits (Table 4; Figure 1).

In samples from the CH group, foamy cells formed layers occupying from a small portion to as much as 75% percent of the vessel circumference. In the W group, the foamy cell deposits typically occupied 33–50% of the vessel circumference. The aortic sections obtained from rabbits receiving Linola flaxseed were characterized by deposits of foamy cells occupying 20–25% of the vessel circumference.

### 3.3. Aorta Contractility

The contraction response to the 60 mM KCl solution was stronger in the CH group compared with the C group. In the L group, the aortic contraction to the high potassium solution was weaker as compared to the CH group. In W group, no statistical differences between the CH, C, and L group were observed (Table 5).

The amplitude of the maximum contractile response to PHE was enhanced by the high-cholesterol diet. However, neither W86 nor Linola flaxseed had a significant influence on the maximum response to PHE (Table 5). After preincubation of the tissues in 10 µM L-NAME, the E_max_ to PHE was still significantly stronger in the CH group compared with the control group. Aortic rings from the L group tended to respond weaker than the aortae from the CH group after preincubation with 10 μM L-NAME. Preincubation of the aortic rings in 100 μM L-NAME ameliorated the differences in the E_max_ among groups (Table 5). 

Differences in the contraction strength to individual concentrations of agonist are shown in Figure 2. Pretreatment of the aortic rings with 100 μM L-NAME resulted in a strong tendency toward an increased maximum reaction amplitude in the C group (*p* = 0.0559). The PHE potency was comparable in all groups irrespective of the L-NAME preincubation (Table 5). No differences in the relaxant responses to SNP were detected among groups (Table 5, Figure 3).

## 4. Discussion

### 4.1. Plasma Lipid Profile

Eleven weeks of feeding the rabbits a 1% cholesterol diet resulted in significantly elevated concentrations of plasma TChol and, specifically, the LDL cholesterol fraction. Neither Linola nor W86 flaxseed influenced plasma lipid parameters. There are conflicting data on the ability of flaxseed and ALA to affect the blood lipid profiles of experimental hypercholesterolemic animals and human patients. The goal of breeding a low in ALA Linola cultivar was to obtain a flax variety appropriate to produce unsusceptible to oxidation oil [18]. The effect reducing serum TChol level was observed in healthy women consuming Linola seed oil for 4 weeks [31]. 

Additionally, Linola flaxseed cakes were found to reduce hypercholesterolemia in mice [21]. Prasad et al. [32,33] demonstrated that classic ALA-rich flaxseed supplementation to rabbits fed a 1% cholesterol diet for 4 and 8 weeks failed to show lower serum TChol and TG concentrations. Crop Development Centre (CDC)-flaxseed with very low ALA (2–3% by mass) increased the TG levels after 4 and 8 weeks in these animals. However, the TChol and LDL cholesterol concentrations were significantly lower in rabbits receiving CDC-flaxseed after 4 weeks. 

The lack of influence of either flaxseed variety in the diet of hypercholesterolemic rabbits on the plasma lipid profile in our study may be associated with a longer duration of the experiment, as in the study on CDC-flaxseed, the cholesterol-lowering effect of the flaxseed tended to diminish with time [32]. However, CDC-flaxseed was more efficient in reducing the serum TChol and LDL concentrations than the ordinary, rich in ALA flaxseed [34].

In our study, a slight upward trend in the plasma TG concentration was seen in the L group. In another study on ALA-rich flaxseed in hypercholesterolemic rabbits, no differences in the serum TChol were detected after 6, 8, or 16 weeks. However, the serum TG concentrations were significantly lower after dietary enrichment with ALA-rich flaxseed compared to rabbits fed only a high-cholesterol diet throughout the experiment [35]. Again, in our study, a tendency toward a decreasing plasma TG concentration was present in the W86 group fed genetically modified Linola with a high ALA content. In humans, flaxseed has been proven to decrease TChol and LDL-cholesterol concentrations by 1.6–18%. In most trials, flaxseed had no effect on the serum TG and HDL-cholesterol concentrations. However, other studies showed no beneficial effects of flaxseed and, rather, an increase in TG levels after flaxseed supplementation [36]. 

### 4.2. Histopathology

Genetically modified flaxseed with a stable high ALA content, protected from oxidation due to increased antioxidant capacity (18), did not inhibit atherosclerosis progression in our rabbits. The dimensions of the foamy cell depositions in the thoracic aorta after dietary enrichment with W86 flaxseed were similar to those of aortas from high-cholesterol diet rabbits; however, low ALA flaxseed proved to slightly inhibit atherogenesis. Both results are consistent with the previously published data [33,35]. 

The lack of a beneficial influence of ALA-rich flaxseed on atherosclerosis progression may be associated with a long duration of cholesterol feeding. Researchers observed that the prolonged administration of cholesterol overwhelmed the inhibitory effects of classic high-ALA flaxseed on atherogenesis in rabbits [35]. However, dietary enrichment with flaxseed containing high amounts of ALA may be beneficial in combination with a change in diet [37].

The higher antioxidant capacity of W86 flax potentially inhibits the oxidation of unsaturated fatty acids in the seeds, which could reduce atheromatic lesions formation, in opposition to the previous study using flaxseed oil with weakly oxidized ALA and LA, which promote changes increasing the oxidative stress and atherogenic risk in LDLr(−/−) mice [20]. Nevertheless, the results of our study indicate that flaxseed with a high ALA content, regardless of number of antioxidants, was less effective at suppressing atherosclerosis progression than low-ALA flaxseed in a high-cholesterol diet. 

This confirmed a previous study that proved a more beneficial effect of CDC-flaxseed that was low in ALA on the lipid metabolism compared to ordinary flaxseed [32,34]. Moreover, ALA in the body mainly undergoes β-oxidation, while only a small part is involved in the synthesis of DHA/EPA, which are thought to have anti-atherosclerotic properties [38,39]. On the other hand, a high ALA concentration may limit LA desaturation, thus, promoting oxidation-susceptible LA incorporation into LDL-cholesterol [40].

The changes in the arterial wall caused by dyslipidemia lead to vascular inflammation and arterial stiffness depending on several factors, such as nitric oxide synthase (eNOS) activity or cholesterol efflux. Previous studies indicated that the long-chain unsaturated fatty acids (UFAs)—but not SFAs—may suppress both apoA-I and HDL-mediated cholesterol efflux from murine macrophages in the artery wall by decreasing the ABCA1 and ABCG1 (ATP binding cassette transporters A1 and G1) gene expression [41,42,43]. 

Thus, as cholesterol efflux is considered to have protective effect on the development of arterial stiffness, it is possible that high levels of UFAs in the artery wall promote atherogenic changes [26]. This phenomenon may explain the lack of protective effect of UFA-rich flaxseed on arterial lesion formation in our rabbits. However, the subsequent study showed that, among different fatty acids, only ALA increased the level of cholesterol efflux in the foam cells [44]. The opposite effect was indicated for LA [45]. W86 seeds have a much lower LA/ALA ratio as compared to Linola seeds, but they did not decrease the aortic lesion progression in the present study.

An in vitro study on endothelial cells confirmed the beneficial effect of a low LA/ALA ratio (1:1) on cardiovascular risk but only at a low PUFA concentration, whereas the same low LA/ALA ratio might raise the risk of cardiovascular diseases at a high PUFA concentration [46]. The effects of LA/ALA ratios on endothelial function might depend on proinflammatory factors, since a low LA/ALA ratio upregulated the nuclear factor κB (a key factor in inflammatory response in atherosclerosis) gene at a high PUFA concentration [46]. Therefore, the high PUFA concentration provided to rabbits with flaxseed-rich diets in our study might change the effect of the low LA/ALA ratio in W86 flaxseed to promote atherosclerosis via enhancing the inflammatory response. Another in vitro study confirmed the anti-inflammatory effects of EPA and DHA but not of plant-derived n-3 PUFAs, including ALA, which is poorly converted into EPA or DHA in the body [12]. 

Considering the effectiveness of low-ALA flaxseed as opposed to genetically modified seeds with high-ALA content, substances other than ALA may be responsible for the beneficial properties of flaxseed. The flax–lignan complex, particularly secoisolariciresinol diglucoside (SDG), has demonstrated anti-atherogenic and anti-inflammatory activities [36]. SDG in doses of 20 mg/kg caused decreases in the serum TChol, TG, and very low-density lipoprotein in mice with experimental hyperlipidemia [47]. The atherogenic changes induced by hypercholesterolemia in the rabbit model were more effectively reduced by SDG compared with the flaxseed intake with no effect of flax oil rich in PUFAs [48]. 

Polyphenols are known to exhibit anti-atherosclerotic properties by preventing the accumulation of cholesterol in macrophages due to the inhibition of lipoprotein oxidation and influence on cholesterol uptake and efflux [49]. However, cakes of genetically modified GT#4 flaxseed with increased contents of SDG, as well as kaempferol, quercetin glycosides, proanthocyanins, and ferulic acid showed similar effect to cakes of Linola seeds on the reduction in TChol and TG in hyperlipidemic mice [21]. In addition, SDG, Linola and W86 flaxseeds are excellent sources of anthocyanins, flavonols, and proanthocyanins; however, Linola W86 is much richer in hydrolysable tannin.

Research indicated that the tannin contained in persimmon promoted macrophage reverse cholesterol transport, which might relieve the process of atherosclerosis [50]. However, in the present study, no protective effect of rich in hydrolysable tannin W86 flaxseed on atherosclerosis progression was observed. This could be related to the multidirectional actions of the hydrolysable tannin, which may also cause mild disturbances in liver and kidney functions following its long-time oral administration [51]. Hepatic dysfunction could cause dyslipidemia resulting in the development of atherosclerotic lesions.

### 4.3. Aorta Contractility

Hypercholesterolemia resulted in significantly increased vascular responses to 60 mM KCl and to PHE in our study. Increased contractions evoked by KCl were also observed in aortae from apoE-/LDLR-mice [52,53]. Studies demonstrated that even a modest elevation in serum cholesterol enhancesthe aorta sensitivity to KCl and phenylephrine [52,53,54]. Researchers postulated that one of the mechanisms of an increased contractile responses in hypercholesterolemia is related to an increased vascular Ca^2+^ load and greater calcium ions influx in vascular smooth muscles after activation [52,53]. Another postulated mechanism of increased contractile reactivity is endothelial dysfunction caused by a decreased NO synthesis and bioavailability, which may be due to proinflammatory state and oxidative stress [52,53,54,55,56]. Low-ALA flaxseed effectively decreased the response to 60 mM KCl in the atherosclerotic rabbit aortae in our study. High-ALA genetically modified flaxseed also caused a less pronounced reduction in the KCl-induced contractions. Both types of flaxseed in our study tended to ameliorate the increased contractile response to phenylephrine due to hypercholesterolemia.

Similar to our results, it was found that flaxseed extract directly decreased the reaction to KCl in isolated rabbit jejunum. A response to the increasing concentrations of calcium was also decreased. The flaxseed extract influence on intestinal smooth muscles was comparable to that of verapamil—a calcium blocker [57]. These results may indicate that flaxseed, regardless of the ALA content, has a positive influence on the vascular smooth muscle cells.

In our study, L-NAME, a nonspecific nitric oxide synthase inhibitor, caused a tendency to increase the maximum response to PHE in the aortic rings from control rabbits. Moreover, a difference in the maximum response to PHE in the C and CH groups was diminished after preincubation with 100 μM of L-NAME. This may be explained by an inhibition of basal endothelial NO synthesis in healthy aortae by L-NAME and a decreased eNOS activity in hypercholesterolemic animals [58,59]. Supplementation of the hypercholesterolemic rabbits with the Linola variety or W86 flaxseed did not result in an increased response to PHE after L-NAME. Therefore, it seems that neither of the supplements protected eNOS activity in the aorta. In aortae from rabbits given Linola, the response to PHE after preincubation with 10 μM L-NAME had a strong tendency to be reduced as compared to tissues from the CH group. One of the possible mechanisms for this effect of Linola is the inhibition of iNOS in the aorta, which is believed to be induced in atherosclerosis. Research demonstrated thatSDG derived from flaxseed inhibited renal iNOS expression in rats with streptozotocin-induced diabetes who were fed a high-fat diet [60]. However, there are some results suggesting that flaxseed oil increases inflammation and ROS activation in the vessel walls [61]. In the case of oxidative stress, L-NAME may, surprisingly, lead to an increased generation of NO in the vessels and macrophages [62], which may explain the lack of increased response to phenylephrine after L-NAME, particularly in the W86 group, in our study. 

The mentioned in vitro study involving endothelial cells indicated different effects of the LA/ALA ratios on the NO level, depending on PUFA concentration, with no effect on the endothelin-1 (ET-1) level [46]. Although a low LA/ALA ratio raised the eNOS gene expression, the same low LA/ALA ratio caused a decrease in the NO/ET-1 and NO levels compared to a higher LA/ALA ratio at a high PUFA concentration. Thus, the flaxseed rich in PUFAs supplied to rabbits for an extended period of time in our experiment might have downregulated the production of NO, especially in case of W86 with a low LA/ALA ratio. 

The suppression of eNOS activity and NO production may be caused by fatty acids—another mechanism of vessel stiffening [26]. Research found that linoleic and oleic acids caused decreases in eNOS activity, whereas elaidic and stearic acids had no effect [63]. Oleic acid also inhibited the endothelium-dependent vasodilator response to acetylcholine in the rabbit femoral artery rings pre-constricted with PHE [63]. Since LA and oleic acid constituted a considerable share in the pool of UFAs in our flaxseeds, their effect on aortic stiffness via decreased eNOS activity in our study was likely. On the other hand, aortic stiffness could be influenced by cholesterol efflux via its interaction with the fatty acid metabolism. 

It is likely that the total amount of fat supplied is of great importance. Protective effects of UFAs on atherogenic changes were obtained using diets low in SFAs [64], but in the case of a high cholesterol diet, the protective effect of these acids may have been counteracted by the above-described actions of UFAs on the stiffening of the arteries. Considering the ambiguous data concerning the UFA impact on the cholesterol efflux and arterial stiffness, some scientists suggested that a decrease in the total fatty acid supply may produce a better protective effect on the arterial wall compared with the changes in the fatty acid composition [42]. Thus, the cumulative effect of a high cholesterol diet together with an elevated level of UFAs could be adverse and might not ameliorate atherogenic change progression.

Neither a high-cholesterol diet nor flaxseed supplementation, regardless of ALA content affected relaxation of the rabbit aorta in response to SNP in our study, which is consistent with the results of Francis et al. [37], Dupasquier et al. [35] and Davda et al. [63].

## 5. Conclusions

Flaxseed with a very low ALA content proved to be more effective at inhibiting atherogenesis in hypercholesterolemic rabbits compared with genetically modified W86 flaxseed with high concentrations of ALA, hydrolysable tannin, and proanthocyanids. Neither *Linola usitatissimum* seeds nor W86 seeds in the diet changed the serum lipid profile during the prolonged period of cholesterol feeding, thus, indicating that the atherogenesis-inhibiting action of Linola is not associated with the lowering of the TChol and LDL-cholesterol concentrations. Linola compared with W86 flax tended to be more efficient in ameliorating an increased aortic contractility in hypercholesterolemic rabbits. The mechanism does not seem to be related to nitric oxide. However, more detailed investigations are required to explain the decreased reaction to L-NAME in flax-supplemented animals.

## Figures and Tables

**Figure 1 foods-10-00534-f001:**
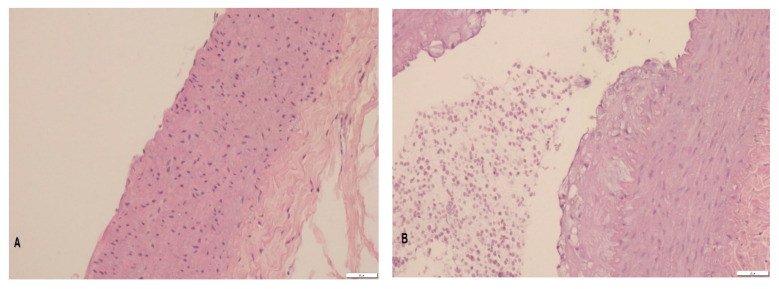
Representative photos showing histological cross-sections of thoracic aortae (hematoxylin-eosin staining; magnification, 200×; bar = 50 µm); (**A**)—control (C) group, normal endothelium and media of the thoracic aorta; (**B**)—CH group, thick layer of atheromatous plaque in the endothelium of the thoracic aorta; (**C**)—W group, layer of atheromatous plaque located in the endothelium of the thoracic aorta composed of foam cells, and disintegrated cellular debris; (**D**)—L group, very thin layer of foamy cells in the endothelium of the thoracic aorta.

**Figure 2 foods-10-00534-f002:**
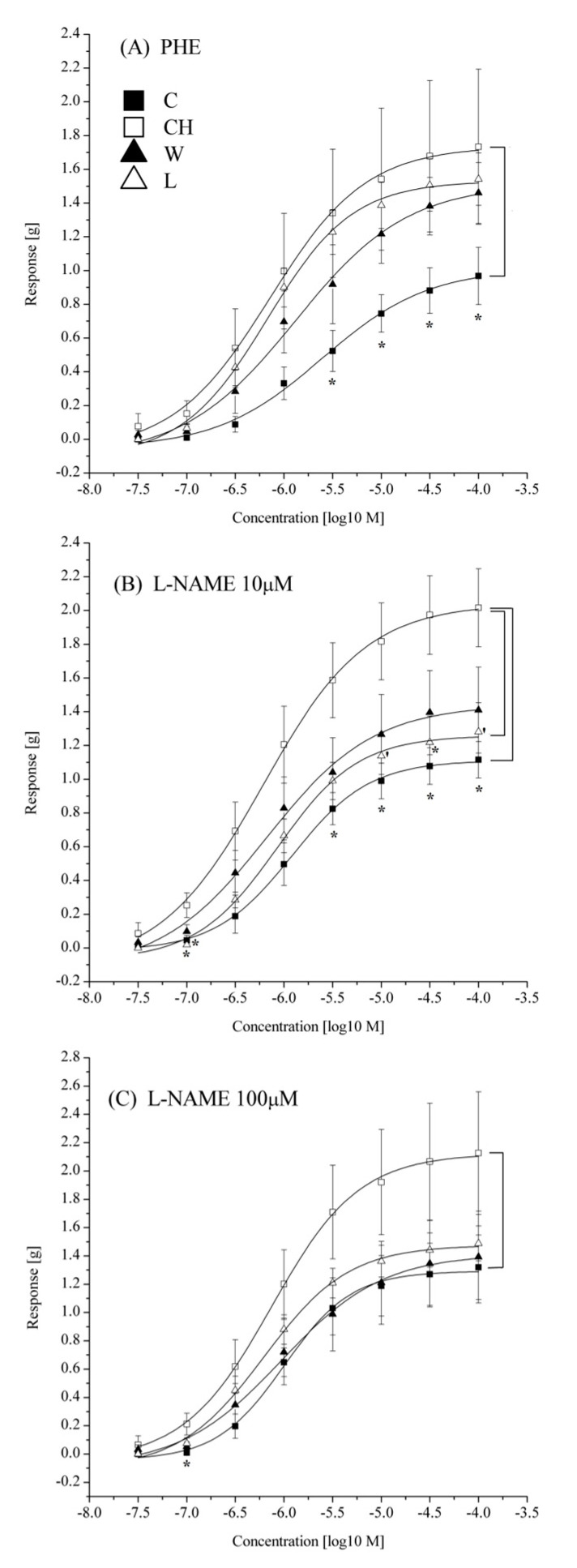
Contractile response of rabbit thoracic aortic rings (**A**) Concentration-response curves to phenylephrine without preincubation; (**B**) after preincubation with 10 μM L-NAME; (**C**) after preincubation with 100 μM L-NAME; C—control group; CH—high-cholesterol diet group; W—W86 flaxseed supplemented group; L—Linola flaxseed supplemented group; PHE—phenylephrine; L-NAME—nitro-L-arginine methyl ester; data are expressed as mean ± S.E.M. of the absolute amplitudes of contraction in grams [g]. * indicate values significantly different from results for CH group at a given concentration (*p* < 0.05); ‘ indicates tendencies (0.05 < *p* < 1).

**Figure 3 foods-10-00534-f003:**
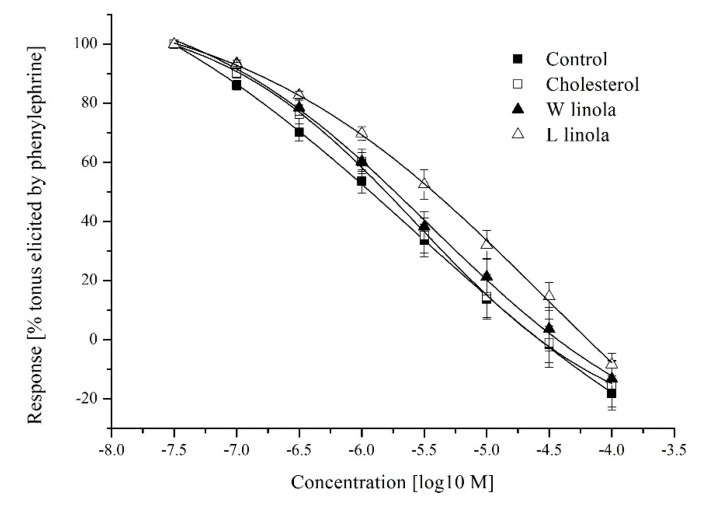
Relaxant response of rabbit thoracic aortic rings to sodium nitroprusside after precontraction with phenylephrine. Data are expressed as mean ± S.E.M. of the relative amplitudes of relaxation in % of contraction elicited by 10^−6^ M PHE.

**Table 1 foods-10-00534-t001:** Fatty acid content in seeds.

Item (μg/gFW)	Linola	W86
16:0	11.33 ± 0.5	13.60 ± 0.5
16:1	0.16 ± 0.02	0.22 ± 0.03
16:2	0.12 ± 0.02	0.17 ± 0.02
16:3 (n-3 FA)	0.10 ± 0.01	0.10 ± 0.01
18:0	6.38 ± 0.25	7.63 ± 0.25
18:1(OA, n-9 FA)	32.57 ± 0.75	40.05 ± 0.68
18:2 (LA, n-6 FA)	158.57 ± 3.85	153.91 ± 4.55
18:3 (ALA, n-3 FA)	3.64 ± 0.5	105.70 ± 2.56
20:0	0.26 ± 0.01	0.26 ± 0.01
20:1(n-9 FA)	0.19 ± 0.01	0.19 ± 0.01
22:0	0.12 ± 0.01	0.14 ± 0.01
22:1(n-9 FA)	0.10 ± 0.01	0.18 ± 0.02
SFA	18.13 ± 0.97	21.70 ± 1.05
MUFA	33.02 ± 0.76	40.64 ± 0.73
PUFA	162.21 ± 2.86	259.61 ± 3.45
n-6/n-3	43/1	1.5/1
n-9/n-6/n-3	9/43/1	1/4/2.5
Total	213.57 ± 3.91	322.22 ± 3.25

FW—Fresh weight, OA—oleic acid, n-9 FA-n-9 fatty acid, LA—linoleic acid, n-6 FA-n-6 fatty acid, ALA—alpha-linolenic acid, n-3 FA-n-3 fatty acid, SFA—saturated fatty acids, MUFA—monounsaturated fatty acids, PUFA—polyunsaturated fatty acids.

**Table 2 foods-10-00534-t002:** Phenolic compounds content in seeds.

Item (mg/gFW)	Linola	W86
Ferulic acid and glucoside	2.131 ± 0.048	2.382 ± 0.043
Coumaric acid and glucoside	1.400 ± 0.059	1.626 ± 0.024
Caffeic acid and glucoside	0.782 ± 0.019	1.060 ± 0.092
Phenolic acids (total)	4.313 ± 0.126	5.068 ± 0.159
Vitexin	0.028 ± 0.05	0.016 ± 0.001
Secoisolariciresinol diglucoside (SDG)	13.31 ± 0.387	14.36 ± 0.020
Coniferyl aldehyde	0.002 ± 0.001	0.003 ± 0.001
Proanthocyanidin	0.025 ± 0.002	0.052 ± 0.008
Hydrolysable tannins	0.048 ± 0.001	0.279 ± 0.013

**Table 3 foods-10-00534-t003:** Plasma lipid profiles of rabbits after 11 weeks of feeding the experimental diets.

Parameter		C	CH	W	L
*n*		5	5	4	5
TChol, mmol/L		0.65 ± 0.12 ^a^	13.05 ± 2.33 ^b^	12.17 ± 1.96 ^b^	12.07 ± 2.85 ^b^
HDL, mmol/L		0.47 ± 0.11	0.74 ± 0.08	0.58 ± 0.05	0.61 ± 0.11
LDL, mmol/L		0.18 ± 0.02 ^a^	7.10 ± 1.40 ^b^	6.04 ± 1.11 ^b^	6.42 ± 1.73 ^b^
TG, mmol/L		0.53 ± 0.09	0.91 ± 0.12	0.77 ± 0.10	1.03 ± 0.29

TChol, total plasma cholesterol; HDL, high-density lipoprotein in plasma; LDL, low-density lipoprotein in plasma; TG, total plasma triglycerides; C, control group; CH, high-cholesterol fed group; W, high-cholesterol fed group supplemented with W86 flaxseed; L, high-cholesterol fed group supplemented with Linola flaxseed; *n*, number of rabbits from which tissues were used in contractility and plasma lipid studies. Values marked with different lower-case letters differ significantly (*p* < 0.05) from each other in a row.

**Table 4 foods-10-00534-t004:** Thickness of the foamy cell depositions in the walls of rabbit thoracic aortae.

Group	*n*	Foamy Cell Deposit Thickness [µm]
C	5	1.13 ± 1.13 ^a^
CH	5	22.43 ± 5.16 ^b^
W	4	27.79 ± 6.43 ^b^
L	5	11.75 ± 3.38 ^a,b^

C, control group; CH, high-cholesterol diet group; W, high-cholesterol diet group supplemented with W86 flaxseed; L, high-cholesterol diet group supplemented with Linola flaxseed; *n*—number of rabbits from which tissues were used for histopathological analysis. Values marked with different lower-case letters differ significantly (*p* < 0.05).

**Table 5 foods-10-00534-t005:** Contractile responses of rabbit thoracic aortae to KCl (a), phenylephrine (b) with or without preincubation in nitro-L-arginine methyl ester (10 or 100 µM), and relaxation elicited by sodium nitroprusside (c).

			C	CH	W	L
a	KCl 60 mM	*n*	23	22	19	20
		E_max_ [g]	0.60 ± 0.07 ^a^	1.01 ± 0.12 ^b^	0.75 ± 0.06 ^a,b^	0.70 ± 0.06 ^a^
b	PHE	n	6	5	5	5
		E_max_ [g]	0.97 ± 0.17 ^a^	1.73 ± 0.46 ^b^	1.46 ± 0.18 ^a,b^	1.54 ± 0.16 ^a,b^
		pK	−5.77 ± 0.18	−6.07 ± 0.17	−5.91 ± 0.24	−6.18 ± 0.07
	PHE + L-NAME 10 µM	n	6	5	4	5
		E_max_ [g]	1.11 ± 0.11 ^b^	2.02 ± 0.23 ^a,^^⌠^	1.41 ± 0.25 ^a,b^	1.28 ± 0.17 ^a,b,^^⌠^
		pK	−5.94 ± 0.11	−6.19 ± 0.16	−6.18 ± 0.16	−6.06 ± 0.06
	PHE + L-NAME 100 µM	n	6	6	4	5
		E_max_ [g]	1.32 ± 0.23	2.13 ± 0.43	1.39 ± 0.32	1.49 ± 0.12
		pK	−6.03 ± 0.10	−6.23 ± 0.15	−6.02 ± 0.19	−6.20 ± 0.10
c	SNP	n	20	21	15	19
		E_max_ [%]	−18.1 ± 4.6	−15.3 ± 8.3	−13.2 ± 6.0	−8.3 ± 3.8
		pK	5.65 ± 0.24	4.97 ± 0.69	5.68 ± 0.06	5.22 ± 0.13

C, control group; CH, high-cholesterol diet group; W, W86 flaxseed supplemented group; L, Linola flaxseed supplemented group; PHE, phenylephrine; L-NAME, nitro-L-arginine methyl ester; SNP, sodium nitroprusside; E_max_, maximum reaction; pK, potency; *n*, number of aortic rings used in the contractility study for a given substance. E_max_ is expressed as the absolute value in grams for the highest concentration PHE and the relative value for the highest concentration of SNP in percentage of contraction elicited by 10^−6^ M PHE. pK is expressed as a logarithm of the concentration eliciting 50% of the maximum contraction. Values marked with different lower-case letters differ significantly (*p* < 0.05) from others in a row. ^⌠^ symbol in superscripts in a row indicates values that tend to differ (*p* = 0.0599).

## Data Availability

The data presented in this study are available on request from the corresponding author.

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
