# Peer review of "Atherosclerosis Development and Aortic Contractility in Hypercholesterolemic Rabbits Supplemented with Two Different Flaxseed Varieties"

_foods, 2021, doi:10.3390/foods10030534_

Round 1

Reviewer 1 Report

This is a complementary study where the authors investigated the antiatherogenic effects of the two varieties of flaxseeds, some points should be revised. 

  1. The abbreviation should be explained the first it appears (For example ALA in the abstract)
  2. The authors used 10% transgenic flaxseed W86 or 10% flaxseed Linola, please explain why did you choose this concentration? have you done toxicity tests?
  3. Figure 1 does not have signs to show the differences between the groups, revise
  4. The caption of Figure 3 should be revised.

Author Response

Response to Reviewer

Thank you very much for your kind review and valuable comments. We have modified the manuscript according to your suggestions.

Please find below the detailed answers to your comments.

  1. We have checked and added the explanation of the abbreviations at their first appearance in the text.
  2. In our project, when selecting the amount of flax to be added to the diet, we used the information available in the literature on flax additives used in experimental animals [1,2]. Moreover, the United States. Food and Drug Agency authorized GRAS status (generally regarded as safe) for whole and ground flaxseed up to 12% (by weight) in the food products [3]. Unfortunately, we did not conduct toxicity studies ourselves. 1. Dupasquier CM, Weber AM, Ander BP, et al. Effects of dietary flaxseed on vascular contractile function and atherosclerosis during prolonged hypercholesterolemia in rabbits. Am J Physiol Heart Circ Physiol. 2006;291(6):H2987-96. 2. Ratnayake WMN, Behrens WA, Fischer PWF, et al. Chemical and nutritional studies of flaxseed (variety Linott) in rats. J Nutr Biochem. 1992;3:232–240. 3. Jheimbach LLC, Port Royal VA. Determination of the GRAS status of the addition of whole and milled flaxseed to conventional foods and meat and poultry products. https://wayback.archive-it.org/7993/20170607013434 /https://www.fda.gov/downloads/Food/IngredientsPackagingLabeling/GRAS/NoticeInventory/UCM269248.pdf, last accessed 19 February 2021.
  3. We have revised the photographs in Figure 1 and we chose high-quality pictures that emphasize the differences we have found in the aortae.
  4. We have revised the caption for Figure 3 and we have removed unnecessary descriptions.

We hope that the improvements to our manuscript that you suggested have made it suitable for publication in Foods.

Reviewer 2 Report

The resent paper evaluate if the Linola flaxseed with a very low ALA content influences serum lipid profile, atherosclerosis progression, and vascular reactivity in hypercholesterolemic rabbits, and elucidate if an increase in stable ALA content in genetically modified seeds would further impact these effects.

Major comments

The aim of the work is clear and the design appropriate, but the description must be improved, and mainly the introduction and discussion must be updated, since bibliography used in the work is obsolete and recent work are publish on the field.

Abstract

Description of the aim in the abstract is not clear. Please clarify this issue.

Introduction

The bibliography used in the manuscript is obsolete and must be updated. There are many recent works, even reviews, that can be used. It may help to a better understanding of the actual information and controversy related to the real effect of ALA and flaxseed on cardiovascular events, and thus justify the aim of the resent work. I would suggest to reduce the description of the first part related to atherolsclerotic plaque formation and better explain the impact of ALA supplementation on cardiovascular health.

Material and Methods

Line 163: please explain the significance of all parameters related with contractile experiments.

Results

Line 190-196 and Table 4: Please clarify information in table 4: describe units. In addition, the description of histopathology does not fit with information in table 4. Please describe or specify properly.

Line 215-220: you describe differences between CH and the others groups but you don´t indicate that differences in the Figure 2. Please check and indicate it properly when statistical differences (p<0.05) or tendencies occurs.

Line 224: Which is Table III??

Table 5: this table is not clear and need to be clarified, maybe the distribution need to be changed.

There are many differences between n values. Please explain these differences.  

Letters (a), (b) and (c) are confusing, maybe because of the black lines or because of the situation of the fist column. Please change it

As described in the methodology section, 60mM is the concentration of KCl, but in this table it is difficult to understand

Line 282-285: Please explain “low ALA flaxseed proved to slightly inhibit atherogenesis… The lack of a beneficial influence of ALA-rich flaxseed on atherosclerosis progression may be associated with a long duration of cholesterol feeding” How do you explain the effect with low-ALA but no effect with High-ALA if both groups of animal have been fed the same cholesterol-rich diet??

Figure 2 needs statistical indications

Discussion

The discussion must be updated in all sections in order to improve the quality of the work

The hipo-cholesterolemic effect of linola is not well discussed. References used are obsolete please use recent works to discuss the results

Line 259: describe the meaning of “CDC-flaxseed”

For example, there are other genetically modified types of flaxseeds (W92) that have demonstrated hipoholesterolemic effects. You may include that in the discussion

Line 279: Please specified what you mean with “protected from oxidation”

Line 291: Please specified what you mean with “which could oppose the antiatherogenic effects”

Lines 319-321: “The widely known anti-atherogenic properties of polyphenols are usually attributed to the ability to regulate cholesterol metabolism [37].” Please revise this information and update

You talk about oxidative stress through the discussion but you did not measure any biomarker of oxidative stress. I would recommend to focus on your results and try to update the discussion of them, to improve the paper.

Minor comments

Line 93. “After acclimatization,” Please specify how many days

Lines 93-96: “The control animals (C) received the basal diet used in the adaptation period. In the cholesterol group (CH), the rabbits were fed a 1% cholesterol-enriched basal diet. In the W and L groups, the rabbits were fed a 1% cholesterol diet 95 containing 10% transgenic flaxseed W86 or 10% flaxseed Linola, respectively”. Please explain it better, for example “The control animals (C) received the basal diet used during the adaptation period. A group of animal was feed a 1% cholesterol-enriched basal diet, and was named the cholesterol group (CH). Two groups of rabbits were feed the same 1% cholesterol-enriched diet containing 10% transgenic flaxseed W86 (group W) or 10% flaxseed Linola, (group L)”

Line 107-108: Join the sentence “The details concerning plant transformation, selection and transgenic plant analysis were described previously [20].” With that on line 116-117 ”Chemical characteristics of the W86 flaxseed were described in detail by Å»uk et al. [20]” , as: “The details concerning plant transformation, selection and transgenic plant analysis, and chemical characteristics of the W86 flaxseed were described in detail previously [20].”

Line 114: replace “Consequently,” with “In addition,”

Table 1: please indicate which fatty acids are n-3, n-6 or n-9, and also differentiate total N-6 PUFA and n-PUFA.

Line 127: replace “the animals were sampled for the plasma lipid profile determination. Total cholesterol (TChol), high-density lipoproteins (HDL), LDL, and triglycerides (TG) 128 in blood plasma were measured” with “the animals were bled, and plasma was isolated by centrifugation at XXX rpm xx min, 4ºC. Total plasma cholesterol (TChol), high-density lipoproteins (HDL), LDL, and triglycerides (TG) were determined ….”

Line 128: if you describe HDL you should also descried LDL the first time it appears. Please, check the journal author´s instructions to see if it is necessary to describe these terms.

Line 133: replace “After 11 weeks,” with “after bleeding,”

Line 150-163: please, include a reference for this methodology.

Line 166: include “n samples specified where applicable”.

Line 175: replace “A high-cholesterol diet for 11 weeks resulted in an elevated blood plasma TChol level. Neither W86 flaxseed nor Linola flaxseed supplementation lowered plasma TChol concentrations. When HDL concentrations remained unchanged in hypercholesterolemic 177 animals, LDL concentrations increased. Mean concentrations of TG were elevated in rabbits fed a high-cholesterol diet regardless of flaxseed supplementation, but the difference was not significant (Table 3)” with “Feeding a high-cholesterol diet for 11 weeks resulted in an elevated blood plasma TChol and LDL levels. Neither W86 flaxseed nor Linola flaxseed supplementation lowered plasma TChol  and LDL concentrations. HDL and TG concentrations remained unchanged in cholesterol-feed animals (Table 3)”. I suggest this later change since no statistical differences are indicated for TG in table 3, but, leases check.

Line 211: replace “The response to 60 mM” with “The contraction response to 60 mM”

Line 253-254: replace “Eleven weeks of being fed a 1% cholesterol diet resulted in significantly elevated concentrations of plasma TChol and LDL cholesterol fraction.” with “Eleven weeks of feeding a 1% cholesterol diet resulted in significantly elevated concentrations of plasma TChol and specifically the LDL cholesterol fraction.”

English language must be revised through the text by a language edited expert

Author Response

Response to Reviewer

Thank you very much for your constructive and careful review and valuable comments. We have modified the manuscript according to your suggestions.

Response to Reviewer 2

Thank you very much for your constructive and careful review and valuable comments. We have modified the manuscript according to your suggestions.

Please find below the detailed answers to your comments.

  1. Abstract

We have changed the description of the aim of the study in the Abstract section. We hope it is more precise now: “Alpha-linolenic acid (ALA) is widely regarded as the main beneficial component of flax for the prevention of cardiovascular disease. We evaluated the effect of the transgenic flaxseed W86 rich in ALA on lipid profile, atherosclerosis progression and vascular reactivity in hypercholesterolemic rabbits compared to the parental cultivar.”

  1. Introduction

We have followed the Reviewer’s comments on this section and the first part on the mechanisms of atherogenic changes formation has been shortened while the paragraph describing the possible role of ALA in cardiovascular health has been updated.

  1. Material and Methods

Line 163: We have tried to explain the meaning of the measured and calculated parameters in the contractility study by adding the description of the analyzed indices and the features of the substances used.

Now it reads: “After the equilibration period, the tissues were stimulated with 60 mM KCl to determine depolarization mediated contractions. After a washout, 10 or 100 µM of nitric oxide synthase inhibitor Nitro-L-arginine methyl ester (L-NAME) was added to some of the chambers, n samples specified where applicable. Following 15 minutes of incubation, a cumulative concentration-response curve to phenylephrine (PHE) was constructed to measure 1-adrenergic mediated contractility. Potency of the PHE was determined by derivatisation pK value which is a concentration of PHE that produces 50% of maximum response. Efficacy of the agonist was expressed as the maximum response. Moreover, the magnitude of the contractile responses to increasing concentrations of PHE were compared. To examine the relaxation response to sodium nitroprusside (SNP) - a nitric oxide donor, the aortic rings were precontracted with 10-6 M PHE and cumulative concentration-response curves to SNP were constructed.”

  1. Results

Lines 190-196: We have described the units (µm) in Table 4 and adjusted the description of the histopathology so that it  matches with the information in the table.

Now it reads: “In the control group, the normal structure of each layer of the aortic wall was pre-served. The high-cholesterol diet resulted in a significant foamy cell deposition within the walls of the thoracic aorta. The addition of transgenic flax did not prevent the formation of atherogenic deposits, and their thickness was comparable to the changes in the CH group. On the other hand, the addition of parental Linola cultivar was associated with the tendency to decrease the thickness of foamy cell deposits, and the thickness of the plaque in the aortic wall did not differ significantly from the control group, as opposed the aortae of the CH and W rabbits (Table 4; Figure 1). Moreover, foamy cells formed layers occupying from a small portion to as much as 75% percent of the vessel circumference. In the W group, the foamy cell deposits typically occupied 33–50% of the vessel circumference. Aortic sections obtained from rabbits receiving Linola flaxseed were characterized by foamy cells occupying 20–25% of the vessel circumference.”

Lines 215-220: We have added descriptions of the differences in Figure 2. This has also led to the revision of the differences shown in Table 5. There were significant differences marked for the whole curves instead of E max and we have corrected these markings. Accordingly, we have  changed the description of the results. We hope that  the explaination is now clear.. It   reads: “After preincubation of the tissues in 10 µM L-NAME Emax to PHE was significantly stronger in CH group than in control group. Aortic rings from L group tended to respond weaker than aortae from CH group after preincubation with 10 M L-NAME. Preincubation of the aortic rings in 100 M L-NAME ameliorated the differences in Emax among groups. When whole curves were analysed, aortae from rabbits fed high cholesterol reacted stronger to PHE than tissues obtained from C, L, and W groups, both after incubation with 10 and 100 M L-NAME. Differences in contraction strength to individual concentrations of agonist are shown in Figure 3. Pretreatment of the aortic rings with L-NAME resulted in a concentration-dependent tendency toward an increased reaction amplitude in the C and CH groups. PHE potency was comparable in all groups irrespective of L-NAME preincubation (Table 5). No differences in relaxant responses to SNP were detected among groups (Table 5, Figure 3).”

Line 224: We are sorry for this error: there is no Table III, of course. We have corrected the number of the table: Table 5.

Table 5: We have added another column to separate the A, B, C letters. We have added the concentration of KCl used to the substance name and changed 60 mM for Emax [g].In the footnotes, we have added  the explanation that n refers to aortic rings used to study a given substance. These values differ since each ring was stimulated with KCl and subjected to SNP while only some rings were incubated in L-NAME before PHE curve. Moreover, as described in the Material and methods section, some aortic rings had to be discarded from the study due to the lack of reaction to KCl or PHE.

  1. Discussion

The Introduction and Discussion have been  modified and updated on the basis of about 20 new scientific papers. However, some older publications have  not been removed due to  their great scientific importance, e.g. studies by Prasad concerning the anti-atherogenic  effect of flaxseed in the rabbit model.

Line 282-285: The purpose of obtaining low in ALA flaxseed has been  explained in the sentence:

“The goal of breeding low in ALA Linola cultivar was to obtain flax variety appropriate to produce unsusceptible to oxidation oil [18].”

The description of hypocholesterolemic effect of Linola has been  introduced in Discussion (lines 360-362):

“The effect reducing serum TChol level was observed in healthy women consuming Linola seed oil for 4 weeks [31]. Additionally, Linola flaxseed cakes was found to reduce hypercholesterolemia in mice [21].”

The meaning of “CDC-flaxseed” has been  explained as follows: “Crop Development Centre (CDC)-flaxseed with very low ALA (2-3% by mass)” (lines 365-366).

The effects of genetically modified flaxseed on animal lipid profile have been described in the Introduction and Discussion sections:

“Previous studies demonstrated anti-inflammatory and cholesterol-lowering effects of rich in bioactive polyphenols transgenic flaxseed cakes consumed by mice and rats [21, 22, 23]. The study concerning W86 indicated decrease in atherogenic indexes and atherogenic predictors in hypercholesteremic rabbits fed W86 flaxseed, despite of only negligible effect on serum lipid profile [24], which led us to make an attempt to estimate the effect of W86 seeds on the development of atherogenic lesions.” (lines 105-111)

“However, cakes of genetically modified GT#4 flaxseed with increased contents of SDG, as well as kaempferol, quercetin glycosides, proanthocyanins and ferulic acid showed similar effect to cakes of Linola seeds on reduction of TChol and TG in hyperlipidemic mice [21].” (lines 254-257)

Line 279: Please specified what you mean with “protected from oxidation”

It has been explained and this sentence in the current version reads: “Genetically modified flaxseed with a stable high ALA content, protected from oxidation due to increased antioxidant capacity, did not inhibit atherosclerosis progression in our rabbits.” (lines 387-388)

Line 291: Please specified what you mean with “which could oppose the antiatherogenic effects”

The meaning of this phrase has been  specified and the new version of this sentence is:

“The higher antioxidant capacity of W86 flax potentially inhibits the oxidation of unsaturated fatty acids in the seeds which could reduce atheromatic lesions formation, in opposition to the previous study using flaxseed oil with weakly oxidised ALA and LA, which promote changes increasing oxidative stress and atherogenic risk in LDLr(−/−) mice [20].” (lines 399-402)

Lines 319-321: The mechanism of anti-atherogenic polyphenols’ effect has been revised and updated: “Polyphenols are known to exhibit anti-atherosclerotic properties by avoiding accumulation of cholesterol in macrophages due to the inhibition of lipoproteins oxidation and influence on cholesterol uptake and efflux [49].” (lines 452-454)

You talk about oxidative stress through the discussion but you did not measure any biomarker of oxidative stress. I would recommend to focus on your results and try to update the discussion of them, to improve the paper.

We are grateful for this valuable comment. Some sentences concerning oxidative stress have been  removed (lines 71-73, 460-461), only the statements referring to the oxidative stress connected with atherogenic change development have been left unchanged (lines 481-484, 526-529).

Minor comments:

Line 93: We have specified the duration of the  acclimatization period: “After two weeks of acclimatization, the animals were divided into four groups (n = 7 each) fed different diets for 11 weeks.”

Lines 93-96: We have rephrase the sentences according to the reviewer’s suggestions: “The control animals (C) received the basal diet used during the adaptation period. A group of animal was feed a 1% cholesterol-enriched basal diet, and was named the cholesterol group (CH). Two groups of rabbits were fed the same 1% cholesterol-enriched diet containing 10% transgenic flaxseed W86 (group W) or 10% flaxseed Linola, (group L).”

Lines 107-108: We have combined  two sentences indicating the article describing how the transgenic flax was obtained, and now the fragment reads: “The details concerning plant transformation, selection and transgenic plant analysis, and chemical characteristics of the W86 flaxseed were described in detail previously [18].”

Line 114: We have rephrased this fragment of the text and now it reads “Increased accumulation of hydrolysable tannins was the result of partial redirection of substrates from flavonoid biosynthesis (lower activity of chalcone synthase) to other pathways of the phenylpropanoid pathway - which increased the amount of gallic acid of the substrate for the production of hydrolysable tannins. These compounds have the ability to bind iron, which is an activator of fatty acid desaturase (FAD3) [20], and consequently the transgenic plants produced more (20-45%) polyunsaturated fatty acids than the control, mainly α-linolenic acid with better stability due to increased antioxidant capacity (Table 1).”

Table 1: We have indicated n-3, n-6, n-9 fatty acids and the n-6/n-3 ratio.

Line 127: We have rewritten  the sentences according to the reviewer’s comment. Now they read: “the animals were bled, and plasma was isolated by centrifugation at XXX rpm xx min, 4ºC. Total plasma cholesterol (TChol), high-density lipoproteins (HDL), LDL, and triglycerides (TG) were determined using an automated analyser Pentra 400 (Horiba ABX, Japan) and dedicated reagents (Horiba ABX, Japan).”

Line 128: We have already described LDL in the Introduction section and therefore it is not explained here.

Line 133:  As suggested,we have replaced “After 11 weeks,” with “after bleeding.”.

Line 150-163: We have added a new citation to the methodology.

Line 166: We have included the suggested phrase.

Line 175: We have changed the results description according to the reviewer’s comment. Now it reads: “Feeding a high-cholesterol diet for 11 weeks resulted in an elevated blood plasma TChol and LDL levels. Neither W86 flaxseed nor Linola flaxseed supplementation lowered plasma TChol and LDL concentrations. HDL and TG concentrations remained unchanged in cholesterol-feed animals (Table 3).”

Line 211: We have changed the sentence according to the reviewer’s suggestion.

Line 253-254: We have rewritten  the sentence according to the reviewer’s comment. Now it reads: “Eleven weeks of feeding a 1% cholesterol diet resulted in significantly elevated concentrations of plasma TChol and specifically the LDL cholesterol fraction.”

English language has been checked by the MDPI Language Editing Service. Please find enclosed the certificate provided by MDPI.

We are very grateful for the valuable comments which have helped us  to rewrite  our manuscript and will be an inspiration for further research.

Round 2

Reviewer 1 Report

The authors answered the raised comments

Author Response

Response to Reviewer

Thank you for your kind review.

The manuscript will be carefully spell-checked before publication if accepted.

Reviewer 2 Report

The present version has improved, and only minor concerns are needed.

Lines 214-216: “The animals were housed individually at 20 °C in cages with free access to water and a standard rabbit diet for 3 weeks (G-103 pelletized mix for rabbits; Granum Animal Nutrition, Lubanow, Poland). After two weeks of acclimatization”

Did you have the animals housed individually during a total of 5 weeks for acclimatization?? Please, revise this information and make it clear.

Line 227-231: “For the W86 plant transformation, a binary vector containing cDNA from Petunia hybrida was used, encoding chalcone synthase (CHS, EMBL/GenBank database acc. no. X04080) in the sense orientation under the control of the 35S promoter and octopine synthase (OCS) terminator. For W86 plant transformation, the cDNA encoding Petunia hybrida chalcone synthase (CHS database,” The information in lines 227-230 is repeated in lines 230-231. Please, eliminate. Have you publish or patented this methodology previously?  Please, indicate it.

Table 3: Please complete the head of the table “Table 3. Plasma lipid profiles of rabbits after 11 weeks of feeding……….”

Lines 357-358: How do you determine the efficacy of the agonist? Contractile responses parameters must be better explained in methodology and results sections, including Table 5 and Figures 2 and 3.

Lines 913-916: Delete “The lack of protective effect of the rich in UFAs flaxseeds on arterial lesion formation in our rabbits may be connected with the suppressing actions of UFAs on the cholesterol efflux from macrophages in the artery walls. Changes in the vascular reactivity may suggest an influence on the calcium handling in the vessel wall and a lack of protective effect on eNOS activity”. This is not a conclusion of your work, this is a speculation and should not be indicated as a conclusion.

Lines 580-581 and 592: you indicate that “Genetically modified flaxseed with a stable high ALA content, protected from oxidation due to increased antioxidant capacity,” this data must be referenced.

Author Response

Response to Reviewer

Thank you for carefully following the revised manuscript and for constructive comments.

Please find the detailed answers below.

  1. Lines 214-216: “The animals were housed individually at 20 °C in cages with free access to water and a standard rabbit diet for 3 weeks (G-103 pelletized mix for rabbits; Granum Animal Nutrition, Lubanow, Poland). After two weeks of acclimatization”

Did you have the animals housed individually during a total of 5 weeks for acclimatization?? Please, revise this information and make it clear.

We are sorry for this mistake. We always acclimatize animals for not less than two weeks. In this experiments animals were acclimatized for 3 weeks.

Please find the corrected text:

“The animals were housed individually at 20 °C in cages with free access to water and a standard rabbit diet for 3 weeks of acclimatization period (G-103 pelletized mix for rabbits; Granum Animal Nutrition, Lubanow, Poland). Then, the animals were divided into four groups (n = 7 each) and fed different diets for 11 weeks.”

  1. Line 227-231: “For the W86 plant transformation, a binary vector containing cDNA from Petunia hybrida was used, encoding chalcone synthase (CHS, EMBL/GenBank database acc. no. X04080) in the sense orientation under the control of the 35S promoter and octopine synthase (OCS) terminator. For W86 plant transformation, the cDNA encoding Petunia hybrida chalcone synthase (CHS database,” The information in lines 227-230 is repeated in lines 230-231. Please, eliminate. Have you publish or patented this methodology previously? Please, indicate it.

Thank you for this comment. We have deleted the duplicated sentence. This method was extensively described in a cited article [18]. It was not patented. Now this fragment reads:

“The cultivar Linola (Linola usitatissimum L.) and transgenic W86 flaxseeds were obtained from the Department of Genetic Biochemistry, Faculty of Biotechnology, University of Wroclaw, where they were cultivated and where the W86 plants were generated. For W86 plant transformation, the cDNA encoding Petunia hybrida chalcone synthase (CHS database, EMBL/GenBank No. X04080), the 35S promoter and the OCS terminator were used [18].”

  1. Table 3: Please complete the head of the table “Table 3. Plasma lipid profiles of rabbits after 11 weeks of feeding……….”

We have extended the heading of Table 3 and now it reads:

“Table 3. Plasma lipid profile of rabbits after 11 weeks of feeding the experimental diets.”

  1. Lines 357-358: How do you determine the efficacy of the agonist? Contractile responses parameters must be better explained in methodology and results sections, including Table 5 and Figures 2 and 3.

Thank you for this comment. We have we have paraphrased and clarified the text.

Efficacy of the agonists can be expressed as the absolute or relative maximal response elicited by the highest concentration of the agonist still causing response. In the case of SNP, a relative value was used in relation to the previous contractile response to phenylephrine given prior to testing the response to the SNP. For phenylephrine, absolute values were used. These are the pharmacodynamic data evaluated in this type of experiments. However, physiologically reactive tissues / cells are most often not stimulated by maximum agonist concentrations, so the responses produced by lower substance concentrations are also taken into account, as shown in Figures 2 and 3.

Please find below the changed fragments:

“The potency of the PHE was determined by the curve fitting with non-linear regression and the derivatization pK value, which is the logarithm of PHE concentration that produces 50% of the maximum response. The efficacy of the agonist was expressed as the maximum response (Emax) to the highest concentration of the agonist still producing an increase in tension. Moreover, the magnitudes of the contractile responses to increasing concentrations of PHE were compared by one-way ANOVA and post-hoc Tukey test and the differences were shown on the graphs. To examine the relaxation response to sodium nitroprusside (SNP)—a nitric oxide donor, the aortic rings were precontracted with 10-6 M PHE, and cumulative concentration–response curves to SNP were constructed. Emax was expressed as percent reaction to 10-6 M phenylephrine elicited by the maximum concentration of SNP. Tissues that failed to produce tension in response to KCl or PHE were discarded.”

“The amplitude of the maximum contractile response to PHE was enhanced by the high-cholesterol diet. However, neither W86 nor Linola flaxseed had a significant influence on the maximum response to PHE (Table 5). After preincubation of the tissues in 10 µM L-NAME, the Emax to PHE was still significantly stronger in the CH group compared with the control group. Aortic rings from the L group tended to respond weaker than the aortae from the CH group after preincubation with 10 mM L-NAME. Preincubation of the aortic rings in 100 mM L-NAME ameliorated the differences in the Emax among groups (Table 5).

Differences in the contraction strength to individual concentrations of agonist are shown in Figure 2. Pretreatment of the aortic rings with 100 mM L-NAME resulted in a strong tendency toward an increased maximum reaction amplitude in the C group (p=0.0559). The PHE potency was comparable in all groups irrespective of the L-NAME preincubation (Table 5). No differences in the relaxant responses to SNP were detected among groups (Table 5, Figure 3).”

Table 5: “Emax is expressed as the absolute value in grams for the highest concentration PHE and the relative value for the highest concentration of SNP in percentage of contraction elicited by 10-6 M PHE.”  

Figure 3: “Data are expressed as mean ± S.E.M. of the relative amplitudes of relaxation in % of contraction elicited by 10-6 M PHE”

  1. Lines 913-916: Delete “The lack of protective effect of the rich in UFAs flaxseeds on arterial lesion formation in our rabbits may be connected with the suppressing actions of UFAs on the cholesterol efflux from macrophages in the artery walls. Changes in the vascular reactivity may suggest an influence on the calcium handling in the vessel wall and a lack of protective effect on eNOS activity”. This is not a conclusion of your work, this is a speculation and should not be indicated as a conclusion.

We have deleted the indicated fragment and replaced it. Now it reads:

“Neither Linola usitatissimum seeds nor W86 seeds in the diet changed the serum lipid profile during the prolonged period of cholesterol feeding, thus, indicating that the atherogenesis-inhibiting action of Linola is not associated with the lowering of the TChol and LDL-cholesterol concentrations. Linola compared with W86 flax tended to be more efficient in ameliorating an increased aortic contractility in hypercholesterolemic rabbits. The mechanism does not seem to be related to nitric oxide. However, more detailed investigations are required to explain the decreased reaction to L-NAME in flax-supplemented animals.”

  1. Lines 580-581 and 592: you indicate that “Genetically modified flaxseed with a stable high ALA content, protected from oxidation due to increased antioxidant capacity,” this data must be referenced.

We have added the reference: 18 (Żuk et al. 2012).

Your in-depth review certainly significantly increased the value of our manuscript. We hope it is now suitable for publication in Foods.
